evolution

environmental variation, heterozygosity, inbreeding, mating system, microsatellite polymorphism, SSR

**Author for correspondence:**
Andrew M. Simons
e-mail: andrew.simons@carleton.ca

# Genotype-environment interaction and the maintenance of genetic variation: an empirical study of *Lobelia inflata* (Campanulaceae)

## Kristen Côté and Andrew M. Simons

Department of Biology, Carleton University, 1125 Colonel By Drive, Ottawa, Ontario, Canada K1S 5B6

(iD) AMS, 0000-0002-0198-465X

High levels of genetic variation are often observed in natural populations, suggesting the action of processes such as frequency-dependent selection, heterozygote advantage and variable selection. However, the maintenance of genetic variation in fitness-related traits remains incompletely explained. The extent of genetic variation in obligately self-fertilizing populations of *Lobelia inflata* (Campanulaceae L.) strongly implies balancing selection. *Lobelia inflata* thus offers an exceptional opportunity for an empirical test of genotype-environment interaction (G × E) as a variance-maintaining mechanism under fluctuating selection: *L. inflata* is monocarpic and reproduces only by seed, facilitating assessment of lifetime fitness; genome-wide homozygosity precludes some mechanisms of balancing selection, and microsatellites are, in effect, genotypic lineage markers. Here, we find support for the temporal G × E hypothesis using a manipulated space-for-time approach across four environments: a field environment, an outdoor experimental plot and two differing growth-chamber environments. High genetic variance was confirmed: 83 field-collected individuals consisted of 45 distinct microsatellite lineages with, on average, 4.5 alleles per locus. Rank-order fitness, measured as lifetime fruit production in 16 replicated multilocus genotypes, changed significantly across environments. Phenotypic differences among microsatellite lineages were detected. Results thus support the G × E hypothesis in principle. However, the evaluation of the effect size of this mechanism and fitness effects of life-history traits will require a long-term study of fluctuating selection on labelled genotypes in the field.

# 1. Introduction

The observation of unexpectedly high levels of enzyme heterozygosity [1,2] and quantitative variation in fitness-related traits in natural populations (reviewed in [3])—has stimulated decades of investigation to explain its maintenance through mutation-selection balance [4] and various forms of balancing selection [5,6]. Proposed variance-maintaining mechanisms include overdominance [7–9], frequency-dependent selection [10–12], antagonistic pleiotropy [13,14], sexually antagonistic selection [15–18] and variable selection in space and/or time [19]. Although the most compelling examples of balancing selection are provided by studies of discrete morphs such as flower colour dimorphisms in angiosperms [7], the relative importance of balancing selection as an explanation of standing genetic variation, in general, remains unknown [20].

Population genetic theory indicates that environmental heterogeneity can maintain additive quantitative genetic variation [21–23], a result supported by some of the few empirical studies on the topic [24–28], but not by others (e.g. [29]). Spatial variation was initially thought to be more effective than temporal in maintaining genetic variation [22], but this was under the assumption of consistent rank order in performance across environments [23]. Under relaxed assumptions about the rank order of fitness across environments—specifically, allowing genotype-environment interaction—temporal heterogeneity can maintain genetic variation [6]. However, standing genetic variation is unlikely to be attributable to a single mechanism (e.g. [28,29]); causal factors may vary among characters, environments and species [29]. Thus, determining which mechanism(s) account for genetic variation in any given context can be a complex task.

Because of several ecological and genetical peculiarities revealed in the recent work on this system, *Lobelia inflata* provides an exceptional opportunity to isolate effects of a single mechanism, namely to perform a test of the hypothesis that temporally variable selection maintains genetic variation. First, a high level of multilocus microsatellite (simple sequence repeats, SSR) variation was observed in two natural populations of *L. inflata*: on average, over two of every five field-collected individuals showed unique genotypic signatures at three or more SSR loci [30]. With a mean of 2.50 alleles per microsatellite locus across 22 loci, the extent of polymorphism is higher than that observed in species with mixed mating systems [31], despite recombination being ineffectual at generating new haplotypes under complete self-fertilization. In this earlier work, genetic lineage was associated consistently with flower colour [30], and significant quantitative genetic variation was found in the timing of flowering [32]. Second, because complete self-fertilization—which is thought to be very rare [33]—leads to zero heterozygosity, this effectively allows the use of microsatellite markers as whole-genome lineage 'labels' [34]. The assumption of complete selfing—based on an anther tube that fully encases the stigma—was supported by microsatellite analysis confirming 100% homozygosity in the Petawawa, Ontario population, which was also used in the present study. Samples from 22 polymorphic loci from five replicate plants from each of 21 maternal lines (i.e. 105 individuals) from the field revealed $F_{IS}$ (inbreeding coefficients [35]) of 1.00 [30], and multilocus SSR alleles are perfectly replicated in every offspring without exception. A third factor is that genetic identity among offspring allows replication of genotypes across manipulated environments; fourth, the species is exclusively sexual and is semelparous, facilitating assessment of lifetime fitness through seed count. Finally, under complete self-fertilization, genome-wide homozygosity and cosexuality, some mechanisms of balancing selection—most definitively heterozygote advantage and sexually antagonistic selection—can be eliminated as alternative explanations.

Frequency-dependent selection is likely to be less important in maintaining genetic variation in selfing than outcrossing populations because no selection acts on mating system components such as through intraspecific competition for pollinators. Likewise, antagonistic pleiotropy—in which a gene influences two or more traits that have opposing effects on fitness—is unlikely to account for maintenance of substantial genetic variation, especially in species characterized by high levels of inbreeding [36,37]. Genetic variation within populations that occurs exclusively among genetic lineages is expected to erode through both amplified effects of genetic drift and selection [38,39] and thus requires explanation.

Of two possible empirical approaches to illuminating variance-maintaining mechanisms, an experimental approach, in which variable selection is imposed on model organisms to observe genetic variation as a response, has most often been taken [24,25,29]. This is a powerful approach, in that the general plausibility of the causal mechanism may be inferred. However, the drawback of this approach is that it cannot ascribe a variance-maintaining mechanism to any particular case in nature; indeed, a major outstanding question is one of the relative importance of the various mechanisms of balancing selection [17]. An alternative would be to adopt an observational approach using a

population characterized by high additive genetic variation and to test a particular hypothesized mechanism under the given ecological scenario [27,28]. The trade-off of an observational approach is that, whereas findings consistent with the hypothesis yield a plausible mechanism operating in nature, the underlying cause cannot be deduced.

Here, we extend the observational approach to include fitness assessments of replicated field-derived microsatellite genotypes across multiple manipulated environments to test the hypothesis that genotype-environment interaction (G × E) in fitness is a plausible explanation for standing genetic variation observed in natural populations of *L. inflata*. The specific prediction consistent with this G × E hypothesis is that fitness ranks of genotypes are environment-dependent; we make no specific predictions about the main effects of environments. Further, we acknowledge that the magnitude and nature of the expression of G × E is necessarily a product of the environments studied and that the effects might differ under a different regime of environmental variation. Although our study could, in theory, be used to infer effects of spatial variation, we posit that temporal variation has overriding influence for several reasons: it has been observed to affect phenotypic expression [40], temporal variation has been shown to cause fluctuating selection on life-history traits [41], the genetic variation occurs over the scale of metres despite limited dispersal ability, habitat conditions are similar across sites, and different populations contain a high frequency of genotypes in common [30]. Whereas a direct test would require a long-term assessment of relative fitness of genetic lineages under naturally fluctuating selection, we instead use a manipulated space-for-time approach in which the fitness of genotyped lineages of *L. inflata* from natural populations is compared across four distinct environments: a field environment, a pseudo-field environment and two contrasting growth-chamber environments. Although more extreme environments might drive stronger G × E, the inference would be more limited than from environments chosen more conservatively as representative of a series of plausible temporal conditions. Still, the inference here is limited to the environment-dependence of genotypic fitness; however, in §4, we present exploratory analyses to suggest whether life-history traits (e.g. time to first flower, time to senescence) are consistently associated with microsatellite lineages, and which are the most likely influences on G × E.

# 2. Materials and methods

## 2.1. Species and field collection

*Lobelia inflata* (Campanulaceae) is a North American native herbaceous monocarpic perennial found in disturbed sandy soils. Seeds germinate from a short-lived seed bank in response to light [42] in spring or following disturbance, and develop into basal vegetative rosettes that have a high probability of overwintering successfully given sufficient snow cover [43]. This rosette, under combined photoperiod and rosette size cues the following spring or summer [44] transitions to a reproductive phase by bolting, or producing a flowering stalk. Individuals typically produce 10–100 or more small perfect flowers in which autogamous self-fertilization occurs. Flowers appear sequentially as the raceme develops and remain open for 3–10 days. The ovaries inflate, turn brown upon senescence, and two valves open on the top of the fruit, allowing passive dispersal by wind or other disturbance [45]. Reproduction is terminal, with no underground storage sufficient for deferred reproduction; although flowers occasionally develop following apparent senescence, these do not contribute to reproductive fitness and are best explained as vestigial artefacts of recent divergence from polycarpic congeners [46].

To ensure sufficient genotypic sample size for the main study, fruit from 83 fully mature *L. inflata* were collected individually within a two-week period in areas surrounding Ottawa, Ontario; 34 from Gatineau Park (Gatineau, Quebec: 45°31′00 N, 75°47′00 W) and 49 from Petawawa Research Forest (Petawawa, Ontario: 45°57′00 N, 77°19′00 W). The 30-year climate normals for Gatineau and Petawawa are similar; for example, average May–August temperatures are 17.70°C and 17.77°C, and average daily maxima are 23.2°C and 23.6°C, respectively (https://climate.weather.gc.ca/climate_normals/index_e.html). In addition to fruit number, height, stem diameter at the base, and the number of branches were measured for each plant at the time of collection; this constitutes the data for the 'field' environment.

Realized sample sizes used in the study were dependent on post hoc microsatellite analyses that reveal replication of genotypes within and across environments (including the field environment); thus, sample sizes are further detailed in the genotyping section (below). For the experimental garden environment, seeds of each of 83 original field individuals were placed under homogeneous growth-chamber conditions (12 h : 12 h, 20°C) in replicate 60 mm Petri dishes lined with moist filter paper until germination. Seedlings were then randomly assigned to 4 cm × 4 cm cells in a 32-celled tray with

autoclaved peat-based soil and transferred into a BioChambers AC-40 growth chamber set at 15 h day : 9 h night photoperiod 24°C : 20°C (a normal or 'medium' season) to induce growth and bolting. Prior to bolting, replicate seedlings derived from each field individual were translocated to random positions within four blocks in an experimental garden (Carleton University: 45°23′00 N, 75°43′00 W) fitted with 30% partial shade cloth and watered as needed. In this small-seeded species, non-genetic environmental effects are expressed predominantly at the intra-parent (i.e. among fruit) level and by individual seedlings via direct microenvironmental effects [47]. Although some proportion of phenotypic variance among field-collected individuals may be caused by microsite variation during development, this individual variation does not confound our analysis both because genotype was randomized with respect to microenvironment across the three manipulated environments, and we are interested in G × E across environments rather than genotype effects expressed within environments. Nonetheless, a set of plants in the medium growth chamber were reserved to act as second-generation seed parents to produce replicated genotypes in the two manipulated 'long-season' and 'short-season' growth-chamber environments.

The same germination protocol was followed in preparing seeds produced in the growth chamber for growth in the long- and short-season environments. For the contrasting chamber environments, germinated seedlings were initially transferred to a 16 h day : 8 h night photoperiod 24°C : 18°C. Upon bolting, however, the plants were split between two chambers. Half of the plants (short chamber) were transferred to a 15 h day : 9 h night photoperiod and 22°C : 16°C thermoperiod, representing a short-cold summer season. The other half of the plants (long chamber) remained under the 16 h day : 8 h night photoperiod and 24°C : 18°C thermoperiod. This 1 h photoperiod difference represents a change in daylength of well over a month and includes dates that plants bolt naturally in the area of study. Each tray was watered 2–3 times per week, and 15 ml of a solution of liquid fertilizer (15-5-15) was added once every two weeks. Final individual fitness, measured as fruit number, was obtained in each environment following fruit maturation and was log-transformed for analyses.

## 2.2. Microsatellite and statistical analyses

Microsatellite genotyping was performed to identify distinct genotypes in the original field-collected individuals and to confirm identity in offspring grown in experimental environments. We followed the protocols outlined in Hughes *et al.* [34]. In addition to using a direct polymerase chain reaction (PCR) protocol where DNA extraction and PCR are combined into a single step, we extracted DNA by clean prep methods using a DNeasy Plant Mini Kit (QIAGEN Inc.) followed by the PCR protocol. Amplification was conducted using a Phire II Direct PCR Kit (Thermo Fisher Scientific), and the PCR was performed in a T-3000 thermocycler (Biometra, Goettingen, Germany). PCR products then underwent high resolution melt (HRM) analysis using SYBR Green protocols in a Rotor-Gene 6000 thermocycler (QIAGEN Inc.) with curve analysis being performed using the Rotor-Gene ScreenClust HRM Software (QIAGEN Inc.). HRM analysis was used to identify single-nucleotide polymorphisms and allele frequencies [48,49]. In earlier work, 22 loci were found to be polymorphic in a single population [34]. However, most loci are redundant despite polymorphism because of linkage (no effective recombination), and this number could be reduced to four informative polymorphic loci in the present study (corresponding to Linflata8, Linflata14, Linflata15 and Linflata21 from [30]), although 13 loci from the field collection were initially genotyped to confirm redundancy. For microsatellite primer sequences, repeat motifs and GenBank accession numbers, see [34].

Fitness was analysed using general linear mixed-effects models, in which environment is fixed, and genotype and G × E effects are random. We are most interested in the G × E factor and, in particular, the component of G × E owing to variance in slopes of genotypic reaction norms across environments—i.e. lack of genetic correlation in breeding values among environments—rather than to environment-specific differences in the magnitude of expression of genotypic variance, or the scale effect (cf. [50–52]). Thus, along with results for the G × E random factor, we report the proportion of this component owing to variance in slopes as a ratio. Specifically, we compare the G × E variance component using a model in which we standardize genotypic variance (genotype mean = 0; s.d. = 1) within each environment to the G × E variance component using original, transformed-only, data. The standard deviation of genotypic means is calculated separately for each analysis to include only the subset of genotypes used across the relevant set of environments. Note that because we use pure inbred lines, the genotypic variance is used as additive genetic variance [53].

The sample size for analyses is dependent on the replication of microsatellite genotypes within the field environment (which was unknown at the time of collection), and on successful propagation and bolting within each of the three experimental environments. This results not only in imbalance but the

variable overlap of nominal genotypes among treatments. We therefore take a hierarchical approach to analysis of fitness across environments, moving from analyses using the highest number of environments (all four), with few replicated genotypes in common (six genotypes), to combinations of fewer environments that include sequentially more genotypes in common (two combinations of three, with 8 and 11 genotypes; two environments with all 16 genotypes in common). Among the 83 field-collected individuals, there were 45 distinct microsatellite genotypes and, of these, 16 were represented by at least two individuals common to at least two environments—including the field environment—used in this study (electronic supplementary material, table S1). Because it is the trade-off between the number of environments and the number of genotypes included that provides the rationale for performing multiple analyses, we do not include combinations of fewer environments if this does not increase the genotypic sample size. Strict application of this hierarchical approach results in four main analyses: six genotypes with replicated individuals occurred in all four environments, 11 replicated genotypes were in common among the long, short and field environments, eight were in common between short, long and garden environments and 16 were in common between the short and long chamber environments, with no other combinations of two environments resulting in an increase in common genotypes. Because we perform multiple tests, we use false discovery rate (FDR) to control for the number of inappropriately rejected null hypotheses [54] for each of these analyses.

Fruit number is assumed here to be a relevant fitness measure (see §1); however, other measures are often used as fitness proxies, and we thus present analyses of the relationships among fruit count and plant height, stem diameter and extent of branching. To confirm the general assumption that variation in microsatellite genotype is associated with variation in phenotype, we analyse sources of variance in flower colour, shown to be consistently associated with genetic lineage in the previous work [30]. In addition, a number of life-history traits, including time from bolting to first flower, height at first flower and time to senescence (first flower to fruit maturation) were measured in the three experimental environments (not measurable at the time of seed collection in the field). Although the focus of this study is on G × E in fitness, and because the presence of genetic variance for fitness components is one of the requirements for maintenance of genetic variance through G × E [55], an analysis of variance of the expression of life-history traits across the two manipulated growth-chamber environments is included.

## 3. Results

The observation of high levels of microsatellite polymorphism in natural populations of *L. inflata* from previous work [34] was confirmed in the present study. There were, on average, 4.5 alleles at each of the four microsatellite loci, revealing 45 distinct genetic lineages among the 83 field-collected individuals. Twenty-two genetic lineages appeared in the 34 individuals collected at Gatineau, and 35 genetic lineages occurred in 49 from Petawawa, with 12 multilocus genotypes shared between the two locations. The 16 unique four-locus homozygous genotypes with sufficient replication to be used in this study showed a mean of 3.0 alleles per locus (electronic supplementary material, table S2). The overall replication of individuals within genotypes used in analyses (electronic supplementary material, table S1) was 6.55 (s.d. = 3.81), and the mean replication of individuals within genotype was consistent for the four analyses, ranging from 6.31 for the analysis of 11 genotypes across three environments to 6.64 for the analysis of 16 genotypes across two environments.

As expected, the manipulated environments generated overall fitness differences: there is a highly significant main effect of environment in all four mixed-model analyses for fruit number after checking for FDR (table 1). According to a post hoc Tukey test, differences occur among every environment, except between the two growth-chamber environments, and only in the two analyses with most reduced genotypic sample size. Although the main effect of genotype is marginally significant according to one of the four analyses—performed on the 16 genotypes present across the two growth-chamber environments—this effect is non-significant ($p = 0.042$) after correcting for multiple comparisons (adjusted FDR critical $p \leq 0.0125$). The factor of main interest, the G × E term, is significant for all four analyses (figure 1), and variance in slopes accounts for, on average, 63% of the G × E variance (table 1). Correlations between final fruit number and fitness proxies are all highly significant (table 2). The partial correlation between fitness (fruit number) and stem diameter is strong, and the correlation between fitness and plant height disappears after controlling for stem diameter and branch number (table 2).

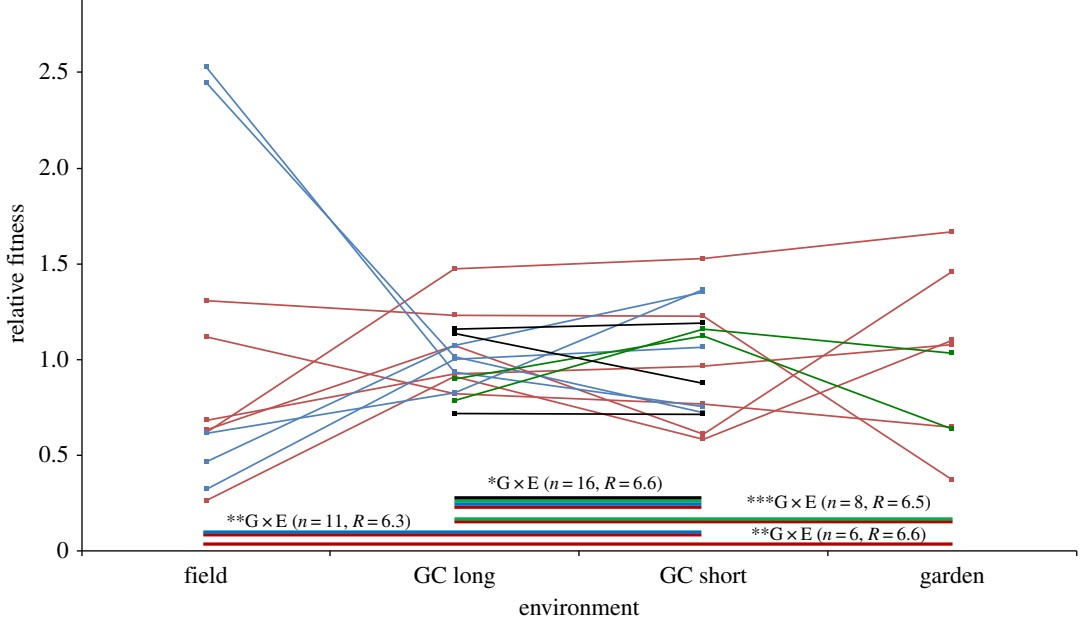

**Figure 1.** Norms of reaction for fitness (fruit number) across the four environments. Fitness is expressed relative to mean fitness (in untransformed units) within each environment (GC, growth chamber). The significance level (* $p \leq 0.05$; ** $p \leq 0.01$; *** $p \leq 0.001$ after accounting for FDR) of the genotype-environment interaction (G × E) terms are given for each combination of environments analysed. The colour bars below the plot indicate subsets of genotypes included in each of the four analyses based on the environments in which they occurred. For example, the six genotypes plotted in red occurred in all environments and are thus included in each of the four analyses. Also shown are genotypic sample sizes ($n$) followed by the mean number of replicates per genotype ($R$). Error bars are omitted for clarity.

**Table 1.** Four mixed-model analyses of variance in fitness as explained by the environment (Env; fixed effect), the genotype (Geno; random effect) and their interaction (G × E; random effect). Analyses were performed for all combinations of environments and genotypes where a decrease in the number of environments results in the inclusion of a greater number of replicated genotypes. VarComp% is the per cent of total variance explained by random factors, and in parentheses is the per cent of the G × E variance component explained by the variance in slopes of norms of reaction (Nrx%). Asterisks indicate significance ($\alpha = 0.05$) after correcting for FDR.

| no. Env, no. Geno | source | MS | d.f. | F | p | VarComp% (Nrx%) |
|---|---|---|---|---|---|---|
| 4, 6 | Env | 7.79 | 3 | 57.6 | <0.0001* | — |
| | Geno | 0.208 | 5 | 1.49 | 0.246 | 3.78 |
| | G × E | 0.148 | 15 | 2.17 | 0.010* | 15.76 (55.0) |
| 3, 8 | Env | 3.25 | 2 | 21.57 | <0.0001* | — |
| | Geno | 0.172 | 7 | 1.065 | 0.433 | 0.74 |
| | G × E | 0.165 | 14 | 2.904 | 0.0007* | 22.25 (78.6) |
| 3, 11 | Env | 9.39 | 2 | 104.4 | <0.0001* | — |
| | Geno | 0.137 | 10 | 1.511 | 0.198 | 5.50 |
| | G × E | 0.099 | 20 | 2.203 | 0.0034* | 16.35 (54.2) |
| 2, 16 | Env | 1.371 | 1 | 30.06 | <0.0001* | — |
| | Geno | 0.124 | 15 | 2.515 | 0.042 | 14.53 |
| | G × E | 0.049 | 15 | 1.989 | 0.025* | 9.07 (63.0) |

The assumption that microsatellite genotype is associated with phenotype is confirmed by the almost perfect consistency in genotypic expression of flower colour ($\chi^2_{30} = 255$; $p < 0.001$), with no evidence of effects of the environment ($\chi^2_2 = 0.0$; $p = 1.0$) or G × E ($\chi^2_{30} = 6.85$; $p = 1.0$) in a nominal logistic regression. Two of the three life-history traits—time to first flower and time to senescence—showed genotypic

**Table 2.** Correlations between fruit number and other fitness proxies. Correlations (restricted maximum-likelihood estimates) above the diagonal (all $p < 0.001$), partial correlations below ($p < 0.05$ except where noted NS).

|  | fruit number | final height | stem diameter | branch number |
|---|---|---|---|---|
| fruit number | — | 0.5589 | 0.7995 | 0.6656 |
| final height | −0.1348 (NS) | — | 0.7106 | 0.5651 |
| stem diameter | 0.6402 | 0.5159 | — | 0.5959 |
| branch number | 0.4109 | 0.2819 | −0.0285 (NS) | — |

**Table 3.** Expression of traits associated with microsatellite genotypes within and across two manipulated experimental environments with maximum genotype sample size. Asterisks indicate significance ($\alpha = 0.05$) after correcting for FDR.

| trait | source | MS | d.f. | F | p |
|---|---|---|---|---|---|
| time to first flower | Env | 1836.63 | 1 | 66.0644 | <0.0001* |
|  | Geno | 108.93 | 15 | 4.2273 | 0.0041* |
|  | G × E | 25.77 | 15 | 0.6594 | 0.8224 |
| height at first flower | Env | 127.25 | 1 | 1.6145 | 0.2207 |
|  | Geno | 93.00 | 15 | 1.0720 | 0.4473 |
|  | G × E | 86.76 | 15 | 2.4643 | 0.0024* |
| time to senescence | Env | 69.80 | 1 | 0.5580 | 0.4617 |
|  | Geno | 613.01 | 15 | 5.5219 | 0.0010* |
|  | G × E | 111.01 | 15 | 0.5464 | 0.9119 |

variation (table 3), with the genotypic expression of height at first flower showing environment-dependence. Only time to first flower differed significantly across the two growth-chamber environments (table 3). The two growth-chamber environments were used for these life-history trait analyses to maximize detection of genetic effects; however, analyses using all three environments in which these traits were measured (eight genotypes in common) yield qualitatively identical results for all effects in all three analyses, with the single exception that the genotypic effect of time to first flower is non-significant.

## 4. Discussion

The evolutionary problem of explaining the maintenance of genetic variation in nature is an enduring one. Theoretical and empirical work has suggested that both spatial and temporal heterogeneity can maintain quantitative genetic variation [21,24–28], although strong negative results [29] imply that mechanisms may be context- and trait-dependent. However, few studies assess the potential role of the underlying mechanism of G × E in the maintenance of quantitative variation through fluctuating selection. Here, we find significant differences in the rank order of genotypic fitness expressed across experimental environments, suggesting the plausibility of G × E as a genetic variance-maintaining mechanism. In agreement with our findings are studies on annual populations of *Mimulus guttatus* that show variable selection acting on flower size [28], as well as on alternative alleles associated with a trade-off between size at, and time to reproduction [26]. In the present study, obligately self-fertilizing, completely homozygous populations of *L. inflata* offer an especially compelling opportunity for a test of mechanisms because inbreeding is expected to affect population genetic structure and lead to high-frequency multilocus genotypes [56]; genetic variance and evolutionary potential are reduced with increasing rates of self-fertilization (e.g. [57]).

The present finding of 45 multilocus microsatellite lineages among the 83 field-collected *L. inflata* individuals is thus unexpected given the mating system. Although interpopulation variance in genetic variation increases with inbreeding [58], our findings do not appear to be a result of sampling anomalous populations in that they are not only consistent with similar recent findings in local populations [30], but

with both microsatellite variation [34] and quantitative genetic variation in life-history traits [32] in three populations (Ontario, Nova Scotia and Massachusetts) of *L. inflata* across northeastern North America. This extent of genetic variation among self-fertilizing lineages thus requires explanation.

Of the differences in rank-order genotypic fitness found across each combination of environments, the strongest G × E effect was expressed across the growth-chamber and the experimental-garden environments (figure 1). For example, the second-highest ranked genotype (CAEA; electronic supplementary material, table S1) in the short growth-chamber environment performed consistently poorly in the experimental garden; the reverse was true of microsatellite genotypes CCEA and ACED, which moved from the two lowest relative fitness rank to second and third highest in the experimental garden. We can infer from our results that not all genotypes express G × E; in particular, one genotype (ACBA; electronic supplementary material, table S1) performed consistently well across all three experimental environments. Interestingly, although it ranked only third in terms of total fruit number under field conditions, this genotype was best represented in the field, with 10 replicates randomly collected, a possible indication of high performance in the previous generation. We note that the irregular availability of specific genotypes under different combinations of environments hampers genotype-specific inference about changes in rank order.

It is difficult to define a natural population when a series of reproductively independent lineages are co-located. For the purposes of our study, we collected genotypes from two main areas surrounding the Ottawa region and, within these areas, individuals are spatially separated on the scale of 1–100 m. Therefore, a caveat in interpreting our results as evidence for the importance of G × E would be that genotypes may be locally adapted on a scale smaller than our collection range, and they do not experience environmental variation to the extent imposed by our four environments. However, this does not appear to be the case: 6 of 16 multilocus genotypes are in common between the two most distant collection locations. Furthermore, microsatellite variation is actually higher within the two most distant collection areas—with 65% and 71% of field-collected individuals being unique genotypes—than when considered as one larger population, in which 54% are unique genotypes. Although all genotypes regardless of collection site were tested under common experimental conditions, it is possible that the G × E results differ across collection sites. We performed an additional analysis in which genotype is nested within the collection site. This analysis included the manipulated environment (E) population of origin (i.e. the site of collection) (Pop), genotype nested within the site of collection (G[Pop]), the genotype-environment interaction (G[Pop] × E) as well as the Pop × E interaction. Although power is reduced (using the dataset with maximum sample size), results confirm a significant effect of the genotype-environment interaction, G[Pop] × E ($p = 0.03$), with no significant effect of the population of origin ($p = 0.88$) or Pop × E interaction ($p = 0.11$). We note, however, that although we frame this study in terms of temporal variation, our results do not preclude spatial G × E as a genetic variance-maintaining mechanism.

We used a series of replicated populations including field-collected and laboratory-grown individuals. Although unlikely in this case (see §2), genotypic effects may have been inflated by parental effects; however, we assess genetic effects across multiple environments rather than within a single environment and, crucially, we are interested in G × E effects rather than main genotype effects. Inflation of G × E effects would not only require that parental effects occur, but that variation in parental effects across environments is genotype-dependent. This is an interesting possibility that, to our knowledge, has not been explored.

Inferences about underlying phenotypic causes of the observed fitness G × E remain hypothetical and are beyond the scope of the present study. However, analyses point to candidate life-history traits underlying environment-specific fitness deserving of future study. The evidence of invariant, 'specialist' traits for particular environment types would include those that show strong genotypic main effects and no rank order change in the trait across environments. Such traits that are optimized for performance in a single environment would thus be candidates for causing fitness variation across environments [59]. By contrast, traits that show weak genotypic effects but significant G × E have potential mitigating effects on changing environments, assuming that the expressed plasticity is adaptive. Here, time to first flower and time to senescence showed strong and relatively fixed genotypic effects, suggesting that they may not contribute to adaptive flexibility in trait expression under changing conditions, and thus may contribute to G × E in fitness. On the other hand, the strong G × E in height at first flower suggests that this trait has the capacity to moderate fitness costs under changed conditions if plasticity in this trait across environments is adaptive.

Although direct assessment of differing fitness effects of underlying life-history traits cannot be made here, an exploratory analysis on the most plausible candidate—height at first flower—indicates its

potential as an underlying but indirect influence on fitness variance. In a two-factor model predicting fitness (fruit number) that includes environment, height at flowering and their interaction, height appears to have a weaker effect ($F = 0.011$; $p = 0.92$) than the interaction ($F = 1.68$; $p = 0.19$). However, in a model that also includes genotype, besides the main effect of environment, G × E is the only significant ($F = 2.15$; $p = 0.014$) predictor of fitness. Thus, the environment-dependent effects of traits on fitness may be complex in that combinations of traits studied may be favoured in particular environments; effects may also be mitigated by traits not included in this study, such as biennial behaviour and variable seed dormancy [41]. Clearly, a targeted study to investigate fitness outcomes of life-history trait expression among genotypic lineages would be needed to test these hypotheses.

Finally, we wish to point out that the selective maintenance of genetic variation in this case—unlike some cases of balancing selection—carries no inference with respect to absolute fitness. It seems reasonable to assume that, given empirical relationships between mating system and inbreeding depression [60] and the very existence of these selfing populations, inbreeding depression may be reduced. However, fluctuating selection maintains genetic variance through its effect on relative—not absolute—fitness, and the population may still be maladapted [61] and either in decline, or maintained through source-sink dynamics thought to be typical of predominantly self-fertilizing populations [58]. Unlike phenotypic plasticity and bet-hedging [62], polymorphism is not considered to be an adaptive response to environmental variation [63].

The characteristics of *L. inflata* that make it well suited for this study (semelparous, reproduces exclusively sexually, obligately self-fertilizing, highly polymorphic) do not themselves moderate the generality of findings; they serve to isolate the potential effects of G × E. Strong effects were observed here using environmental variation that was at least in part artificially generated. We note that the potential importance of G × E is supported in principle by the present results derived from manipulated environments; however, because it is the particular qualities and frequencies of sequential environments that will determine the maintenance of genetic variation in nature, quantitative inferences about the importance of temporal fluctuations in maintaining genetic variation cannot be drawn from the present study. Thus, future study to track fitness of labelled genotypes in the field over the longer term will be required to provide insight into the relative importance of G × E in maintaining genetic variation in natural populations.

Data accessibility. Data are available from the Dryad Digital Repository at https://doi.org/10.5061/dryad.83bk3j9mj [64]. Microsatellite primer sequences and repeat motifs have been deposited in GenBank; accession numbers are listed in Table 1 of [34].

Authors' contributions. A.M.S. conceived of the study; K.C. and A.M.S. designed the study and conducted fieldwork; K.C. conducted all laboratory work including microsatellite analyses; K.C. and A.M.S. performed statistical analyses; K.C. and A.M.S. drafted, revised and approved the paper for submission.

Competing interests. The authors declare no financial or non-financial competing interests.

Funding. This research was supported by the Natural Sciences and Engineering Research Council of Canada (NSERC) Discovery Grant to A.M.S.

Acknowledgements. We thank Dr Jessica Forrest and Dr Naomi Cappuccino for valuable feedback and advice; Hebah Mejbel and Dr Ryan Chlebak for discussion; Dr Susan Aitken for generous logistical laboratory support; Dr P. William Hughes for guidance with genotyping; and four anonymous reviewers for suggestions. We also thank Peter Arbour of the Petawawa Research Forest and Isabelle Beaudoin-Roy of Gatineau Park for coordinating land access and research permits.

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
