## [Reviewer comments · Royal Society Open Science]

Review History

RSOS-191720.R0 (Original submission)

Review form: Reviewer 1

Is the manuscript scientifically sound in its present form?

Yes

Are the interpretations and conclusions justified by the results?

Yes

Is the language acceptable?

Yes

Do you have any ethical concerns with this paper?

No

Have you any concerns about statistical analyses in this paper?

No

Recommendation?

Accept with minor revision (please list in comments)

Comments to the Author(s)

I appreciate the response of the authors to my previous review/comments. Overall I think the manuscript is improved. However, there are a few instances where the additions the authors made to previous comments come across as out of place, taking away from the flow of the paper. I think a little more effort to incorporate those changes into the manuscript would improve the readability, strengthening the manuscript.

- 1) Lines 49-54: Specific, detailed, example of overdominance in a system in the opening paragraph seems a bit random and out of place.
- 2) Lines 94-95: While I appreciate the authors adding this in response to my comment, it reads awkwardly and comes across as a fairly big caveat. I don't think this needs to be the case. Do the authors have any reason to think that pathogens are important in this system? Perhaps rewording this sentence to say something more like, "Frequency dependent selection is likely to be less important in selfing systems than outcrossing ones, as no selection acts on mating systems components ... etc". This way it makes it clear you are not completely negating a potential role for frequency dependent selection, but explaining why it may be less important in selfing systems. Then you could just drop the throwaway pathogen line.
- 3) Line 362: If I am following this argument correctly, time to first flower and time to senescence likely do not contribute to GxE for fitness (as opposed to "thus may contribute to GxE in fitness" as stated in the manuscript), as these two traits "showed strong and relatively fixed genotypic effects". Is this correct? Whereas height at first flower is more likely to contribute to GxE in fitness? Is this is not correct, I am not following the argument made in the paragraph completely, and need further clarification.
- 4) Lines 372-373: Again, I appreciate the authors addressing a comment of mine regarding the potential influence of a seed bank, however, presented at this point in the paragraph/discussion it comes across as a throw away sentence, out of place and not contributing much to the paper. I would either further develop this argument a bit, or get rid of it (or move it).

Review form: Reviewer 2

Is the manuscript scientifically sound in its present form?

Yes

Are the interpretations and conclusions justified by the results?

No

Is the language acceptable?

Yes

Do you have any ethical concerns with this paper?

No

Have you any concerns about statistical analyses in this paper?

No

Recommendation?

Reject

Comments to the Author(s)

Thank you for the work done in clarifying this manuscript. Unfortunately, I still do not think that the conclusions the authors draw match with their results. I do not think that the GxE shown in this study is sufficient to support temporal GxE as a mechanism for maintaining variation in this system. There is some GxE across these manipulated environments, but this in itself not particularly surprising, and does little to support the main hypothesis of the paper.

Studying these genotypes in more clearly stressed conditions in growth chambers - such as drought or cold stress would provide more interesting results across genotypes and better address the hypothesis than the treatments used in this study.

Review form: Reviewer 3 (Melis Akman)

Is the manuscript scientifically sound in its present form?

Yes

Are the interpretations and conclusions justified by the results?

Yes

Is the language acceptable?

Yes

Do you have any ethical concerns with this paper?

No

Have you any concerns about statistical analyses in this paper?

No

Recommendation?

Accept with minor revision (please list in comments)

Comments to the Author(s)

I think the authors addressed my major comments well in the manuscript. I understand the limitation of sample size for the proposed analyses for this dataset. Please see my specific comments below:

Abstract:

24: meaning of "levels" is not very clear. Do you mean variation is continuous, which could be as a result of several mechanisms such as f-dependent selection, heterozygote advantage and/or variable selection?

28: Before *L. inflata* you need a connection word, such as "Thus".

39: Before "Although", the abstract can use a strong statement about the finding of the study.

Intro:

I like the new Intro a lot. I think the authors addressed my suggestions, thanks for that!

128: I do not think you need the sentences after "Each ...". Instead give a brief explanation of the results to engage the readers. I think it is ok to discuss the limitations in Discussion only and not mention them in the Intro.

Mat&Met:

150: Fruits or fruit?

218: Specifically misspelled.

Results and Discussion:

I think the 2nd and 3rd paragraphs in Discussion read more like Results, and would be better suited for that section. I am sorry this is a vague comment but I think the Discussion needs more work with references to earlier studies. Using the framework constructed in the Intro can guide to talk about why this study agrees/conflicts with some other studies or species would add to the Discussion.

Decision letter (RSOS-191720.R0)

04-Nov-2019

Dear Dr Simons,

The editors assigned to your paper ("Genotype-environment interaction and the maintenance of genetic variation: an empirical study of *Lobelia inflata* (Campanulaceae)") have now received comments from reviewers. We would like you to revise your paper in accordance with the referee and Associate Editor suggestions which can be found below (not including confidential reports to the Editor). Please note this decision does not guarantee eventual acceptance.

Please submit a copy of your revised paper before 27-Nov-2019. Please note that the revision deadline will expire at 00.00am on this date. If we do not hear from you within this time then it will be assumed that the paper has been withdrawn. In exceptional circumstances, extensions may be possible if agreed with the Editorial Office in advance. We do not allow multiple rounds of revision so we urge you to make every effort to fully address all of the comments at this stage. If deemed necessary by the Editors, your manuscript will be sent back to one or more of the original reviewers for assessment. If the original reviewers are not available, we may invite new reviewers.

- Data accessibility

<http://datadryad.org/submit?journalID=RSOS&manu=RSOS-191720>

- Competing interests

- Authors' contributions

- Acknowledgements

- Funding statement

Kind regards,

Andrew Dunn

Senior Publishing Editor

on behalf of Prof Kevin Padian (Subject Editor)

Associate Editor's comments:

The reviewers have provided a range of comments and suggestions for further improvements. The most problematic comments (from our perspective, and by implication yours) come from Reviewer 2, who expresses concerns regarding the study conclusions that drawn from your data. Please seriously consider how to address these concerns - you need to persuade the Editors and this reviewer that you've made further sincere efforts to improve the manuscript and delineate changes made both in a point-by-point response and tracked changes manuscript.

Editor comments:

Thanks for your submission. Please attend to all comments carefully; we may not be able to send it out for review again.

Comments to Author:

Reviewers' Comments to Author:

Reviewer: 1

Comments to the Author(s)

I appreciate the response of the authors to my previous review/comments. Overall I think the manuscript is improved. However, there are a few instances where the additions the authors made to previous comments come across as out of place, taking away from the flow of the paper. I think a little more effort to incorporate those changes into the manuscript would improve the readability, strengthening the manuscript.

- 1) Lines 49-54: Specific, detailed, example of overdominance in a system in the opening paragraph seems a bit random and out of place.
- 2) Lines 94-95: While I appreciate the authors adding this in response to my comment, it reads awkwardly and comes across as a fairly big caveat. I don't think this needs to be the case. Do the authors have any reason to think that pathogens are important in this system? Perhaps rewording this sentence to say something more like, "Frequency dependent selection is likely to be less important in selfing systems than outcrossing ones, as no selection acts on mating systems components ... etc". This way it makes it clear you are not completely negating a potential role for frequency dependent selection, but explaining why it may be less important in selfing systems. Then you could just drop the throwaway pathogen line.
- 3) Line 362: If I am following this argument correctly, time to first flower and time to senescence likely do not contribute to GxE for fitness (as opposed to "thus may contribute to GxE in fitness" as stated in the manuscript), as these two traits "showed strong and relatively fixed genotypic effects". Is this correct? Whereas height at first flower is more likely to contribute to GxE in fitness? Is this is not correct, I am not following the argument made in the paragraph completely, and need further clarification.
- 4) Lines 372-373: Again, I appreciate the authors addressing a comment of mine regarding the potential influence of a seed bank, however, presented at this point in the paragraph/discussion it comes across as a throw away sentence, out of place and not contributing much to the paper. I would either further develop this argument a bit, or get rid of it (or move it).

Reviewer: 2

Comments to the Author(s)

Thank you for the work done in clarifying this manuscript. Unfortunately, I still do not think that the conclusions the authors draw match with their results. I do not think that the GxE shown in this study is sufficient to support temporal GxE as a mechanism for maintaining variation in this system. There is some GxE across these manipulated environments, but this in itself not particularly surprising, and does little to support the main hypothesis of the paper.

Studying these genotypes in more clearly stressed conditions in growth chambers - such as drought or cold stress would provide more interesting results across genotypes and better address the hypothesis than the treatments used in this study.

Reviewer: 3

Comments to the Author(s)

I think the authors addressed my major comments well in the manuscript. I understand the limitation of sample size for the proposed analyses for this dataset. Please see my specific comments below:

Abstract:

24: meaning of "levels" is not very clear. Do you mean variation is continuous, which could be as a result of several mechanisms such as f-dependent selection, heterozygote advantage and/or variable selection?

28: Before *L. inflata* you need a connection word, such as "Thus".

39: Before "Although", the abstract can use a strong statement about the finding of the study.

Intro:

I like the new Intro a lot. I think the authors addressed my suggestions, thanks for that!

128: I do not think you need the sentences after "Each ...". Instead give a brief explanation of the results to engage the readers. I think it is ok to discuss the limitations in Discussion only and not mention them in the Intro.

Mat&Met:

150: Fruits or fruit?

218: Specifically misspelled.

Results and Discussion:

I think the 2nd and 3rd paragraphs in Discussion read more like Results, and would be better suited for that section. I am sorry this is a vague comment but I think the Discussion needs more work with references to earlier studies. Using the framework constructed in the Intro can guide to talk about why this study agrees/conflicts with some other studies or species would add to the Discussion.

Author's Response to Decision Letter for (RSOS-191720.R0)

See Appendix A.

RSOS-191720.R1 (Revision)

Review form: Reviewer 1

Is the manuscript scientifically sound in its present form?

Yes

Are the interpretations and conclusions justified by the results?

Yes

Is the language acceptable?

Yes

Do you have any ethical concerns with this paper?

No

Have you any concerns about statistical analyses in this paper?

No

Recommendation?

Accept as is

Comments to the Author(s)

I am happy with the edits made by the authors and think the manuscript is acceptable for publication. The only final point I would suggest is that the authors add a sentence to the first paragraph in the discussion clearly stating the main finding of their paper. This would make the opening of the discussion much more compelling.

Review form: Reviewer 4

Is the manuscript scientifically sound in its present form?

No

Are the interpretations and conclusions justified by the results?

No

Is the language acceptable?

Yes

Do you have any ethical concerns with this paper?

No

Have you any concerns about statistical analyses in this paper?

Yes

Recommendation?

Major revision is needed (please make suggestions in comments)

Comments to the Author(s)

I think that the research explore an important issue in understanding the maintaining of genetic variation by considering GxE (E means environment change in space or time) interaction as a process of natural diversification. However, the analysis of the data is not clear enough, and it

was done in an inefficient manner in order to answer the questions. Conclusions are not clearly justified by a non-efficient analysis.

I think that there are lack of results and more explanation in data analysis. For example, many results presented in the text can be presented more clearly in tables. The authors discussed about the rank of the genotypes but this information is not presented and it is so difficult to read from the figure 1. Model including the region of origin is not described. It is not clear why the authors decided to adjust p-value by FDR sometimes yes and sometimes no. They are not considering that using FDR, they are increasing the rate of false negatives and they use the FDR to argue that genotype effect is not significant. In this specific case, a large probability of error type II is worse than a large probability of error type I. I think that it is necessary to include more tables and a better description of the analysis.

The authors said that they used mixed-effect models, and the best method to estimate variance components is restricted maximum likelihood (REML), however, results in Table 1 match with the results of using minimum least square because they include mean squares. Using REML method allows analyzing all the data together, even with the present unbalance, and it avoids analyzing separately pieces of data that make more difficult to understand the results. Assuming that Geno and GxE are random, the test for genotypic effect has to be built dividing the $MS_{genotypes}$ by MS_{GxE} in the case (3,11), the F value should be $0.137/0.909=0.1507$, in the others three cases values are not exactly the same but they match better.

When analyzing GxE it is possible to separate two components, one due to cross over interaction (COI) and the other one to non-cross over interaction. COI is related with changes in the rank of genotypes and non-COI is related just with changes in the variances. The first kind of interaction is more important than the non-COI, because it explains changes in the behavior of genotypes while the non-COI explains differences in magnitude only. To know how much of both components are in the GxE would be very nice in order to interpret better the results.

As the author mentioned, Figure 1 is not easy to understand because the different data sets analyzed, an analysis using REML and all the data together could clarify the interpretation of results. However, I have to mention that a mixed model will not solve all the problems of unbalance. GxE will always be estimated with available data. X-axis in figure 1 shows the different environments and they can be sorted in many different ways with different effect on the reader. I think that it is better if one graph for each analysis could be less confuse. Others options are: i) to substitute the environment by the average relative fitness of the environment, or ii) to do two-by-two graphs. Graphing by pairs of environments will show more clearly the kind of interaction between the two environments.

When analyzing statistical interaction, we need to consider two environments and two genotypes to measure the level of interaction, then, we cannot talk about the interaction of one genotype or one environment if it is not relative to another set of genotypes and environments.

In summary, I recommend to reanalyze the data, to include more tables and more results and to present them in a better way.

Decision letter (RSOS-191720.R1)

11-Feb-2020

Dear Dr Simons:

On behalf of the Editors, I am pleased to inform you that your Manuscript RSOS-191720.R1

entitled "Genotype-environment interaction and the maintenance of genetic variation: an empirical study of *Lobelia inflata* (Campanulaceae)" has been accepted for publication in Royal Society Open Science subject to minor revision in accordance with the referee suggestions. Please find the referees' comments at the end of this email.

The reviewers and Subject Editor have recommended publication, but also suggest some minor revisions to your manuscript. Therefore, I invite you to respond to the comments and revise your manuscript.

- Ethics statement

- Data accessibility

If you wish to submit your supporting data or code to Dryad (<http://datadryad.org/>), or modify your current submission to dryad, please use the following link:
<http://datadryad.org/submit?journalID=RSOS&manu=RSOS-191720.R1>

- Competing interests

- Authors' contributions

- Acknowledgements

- Funding statement

Because the schedule for publication is very tight, it is a condition of publication that you submit the revised version of your manuscript before 20-Feb-2020. Please note that the revision deadline will expire at 00.00am on this date. If you do not think you will be able to meet this date please let me know immediately.

Kind regards,
Andrew Dunn
Senior Publishing Editor

Royal Society Open Science Editorial Office
 Royal Society Open Science
 openscience@royalsociety.org

on behalf of Prof Kevin Padian (Subject Editor)
 openscience@royalsociety.org

Associate Editor Comments to Author:

Please ensure that you address the final point suggested by the Reviewer 1: "The only final point I would suggest is that the authors add a sentence to the first paragraph in the discussion clearly stating the main finding of their paper. This would make the opening of the discussion much more compelling."

Reviewer 4's comments were solicited to take into account that the more critical reviewer in the previous round of review was unavailable to assess the revision - we apologise this has taken longer than usual. We're of the view that, given earlier reviewers were broadly in favour of publication, we can move toward that goal - though we'd recommend you take the Reviewer 4 comments on board (as far as practical), as they offer a number of potentially useful suggestions for clarifications.

Reviewer comments to Author:

Reviewer: 1

Comments to the Author(s)

I am happy with the edits made by the authors and think the manuscript is acceptable for publication. The only final point I would suggest is that the authors add a sentence to the first paragraph in the discussion clearly stating the main finding of their paper. This would make the opening of the discussion much more compelling.

Reviewer: 4

Comments to the Author(s)

I think that the research explore an important issue in understanding the maintaining of genetic variation by considering G×E (E means environment change in space or time) interaction as a process of natural diversification. However, the analysis of the data is not clear enough, and it was done in an inefficient manner in order to answer the questions. Conclusions are not clearly justified by a non-efficient analysis.

I think that there are lack of results and more explanation in data analysis. For example, many results presented in the text can be presented more clearly in tables. The authors discussed about the rank of the genotypes but this information is not presented and it is so difficult to read from the figure 1. Model including the region of origin is not described. It is not clear why the authors decided to adjust p-value by FDR sometimes yes and sometimes no. They are not considering that using FDR, they are increasing the rate of false negatives and they use the FDR to argue that genotype effect is not significant. In this specific case, a large probability of error type II is worse than a large probability of error type I. I think that it is necessary to include more tables and a better description of the analysis.

The authors said that they used mixed-effect models, and the best method to estimate variance components is restricted maximum likelihood (REML), however, results in Table 1 match with the results of using minimum least square because they include mean squares. Using REML method allows analyzing all the data together, even with the present unbalance, and it avoids analyzing separately pieces of data that make more difficult to understand the results. Assuming that Geno and G×E are random, the test for genotypic effect has to be built dividing the

MS_genotypes by MS_GxE in the case (3,11), the F value should be $0.137/0.909=0.1507$, in the others three cases values are not exactly the same but they match better.

When analyzing GxE it is possible to separate two components, one due to cross over interaction (COI) and the other one to non-cross over interaction. COI is related with changes in the rank of genotypes and non-COI is related just with changes in the variances. The first kind of interaction is more important than the non-COI, because it explains changes in the behavior of genotypes while the non-COI explains differences in magnitude only. To know how much of both components are in the GxE would be very nice in order to interpret better the results.

As the author mentioned, Figure 1 is not easy to understand because the different data sets analyzed, an analysis using REML and all the data together could clarify the interpretation of results. However, I have to mention that a mixed model will not solve all the problems of unbalance. GxE will always be estimated with available data. X-axis in figure 1 shows the different environments and they can be sorted in many different ways with different effect on the reader. I think that it is better if one graph for each analysis could be less confuse. Others options are: i) to substitute the environment by the average relative fitness of the environment, or ii) to do two-by-two graphs. Graphing by pairs of environments will show more clearly the kind of interaction between the two environments.

When analyzing statistical interaction, we need to consider two environments and two genotypes to measure the level of interaction, then, we cannot talk about the interaction of one genotype or one environment if it is not relative to another set of genotypes and environments.

In summary, I recommend to reanalyze the data, to include more tables and more results and to present them in a better way.

Author's Response to Decision Letter for (RSOS-191720.R1)

See Appendix B.

Decision letter (RSOS-191720.R2)

19-Feb-2020

Dear Dr Simons,

It is a pleasure to accept your manuscript entitled "Genotype-environment interaction and the maintenance of genetic variation: an empirical study of *Lobelia inflata* (Campanulaceae)" in its current form for publication in Royal Society Open Science.

Due to rapid publication and an extremely tight schedule, if comments are not received, your paper may experience a delay in publication. Royal Society Open Science operates under a continuous publication model. Your article will be published straight into the next open issue and

this will be the final version of the paper. As such, it can be cited immediately by other researchers. As the issue version of your paper will be the only version to be published I would advise you to check your proofs thoroughly as changes cannot be made once the paper is published.

Kind regards,

on behalf of the Associate Editor, and Professor Kevin Padian (Subject Editor)
openscience@royalsociety.org

Appendix A

November 26, 2019

Dear Drs. Andrew Dunn and Kevin Padian,

Thank you for the opportunity to respond to the reviewers' comments. I include all detailed responses in-line below, in blue font and prefixed by "AMS:" (retaining the reviewers' comments intact) and include a document with tracked changes. Revisions in response to reviewers #1 and #3 were straightforward. As you point out, the most problematic comments are those of Reviewer #2, who clearly remains sceptical about "the study conclusions drawn" from the results, rather than about the statistical validity of our results themselves (significant genotype-environment interactions [GxE] caused by variable slopes). Specifically, the issue is about inferences from observed GxE to maintenance of genetic variation. I completely agree in principle with the criticism. However, I wish to point out that the manuscript explicitly makes a claim that is not as strong as the claim being criticized, and is one that is supported by the results. I have therefore made an earnest effort to respond to this criticism by clarifying hypotheses under test, and by fully acknowledging and explaining limits on inference.

I hope the manuscript revisions are to your satisfaction.

Sincerely,

Andrew M. Simons

Author response to reviewer comments:

Reviewer: 1

Comments to the Author(s)

I appreciate the response of the authors to my previous review/comments. Overall I think the manuscript is improved. However, there are a few instances where the additions the authors made to previous comments come across as out of place, taking away from the flow of the paper. I think a little more effort to incorporate those changes into the manuscript would improve the readability, strengthening the manuscript.

1) Lines 49-54: Specific, detailed, example of overdominance in a system in the opening paragraph seems a bit random and out of place.

AMS: I agree, and details have been removed. We retain only simple mention of the example to bridge from discrete polymorphism to quantitative genetic variation.

2) Lines 94-95: While I appreciate the authors adding this in response to my comment, it reads awkwardly and comes across as a fairly big caveat. I don't think this needs to be the case. Do the authors have any reason to think that pathogens are important in this system? Perhaps

rewording this sentence to say something more like, “Frequency dependent selection is likely to be less important in selfing systems than outcrossing ones, as no selection acts on mating systems components ... etc”. This way it makes it clear you are not completely negating a potential role for frequency dependent selection, but explaining why it may be less important in selfing systems. Then you could just drop the throwaway pathogen line.

AMS: We have re-worded, as requested, so as not to imply that frequency-dependent selection is eliminated as a possibility. Note that support for fluctuating selection as a mechanism does not require elimination of other mechanisms.

3) Line 362: If I am following this argument correctly, time to first flower and time to senescence likely do not contribute to GxE for fitness (as opposed to “thus may contribute to GxE in fitness” as stated in the manuscript), as these two traits “showed strong and relatively fixed genotypic effects”. Is this correct? Whereas height at first flower is more likely to contribute to GxE in fitness? Is this is not correct, I am not following the argument made in the paragraph completely, and need further clarification.

AMS: This was confusing because the distinction between fitness components (time to first flower etc.) and fitness itself (fruit number) as not made clearly enough. In these interpretations in the Discussion (that we were asked to include), we hypothesize that components that are inflexible across environments (i.e. with strong genotypic effects) might be an underlying cause of rank-order differences in fitness itself, following the logic of, for example Sara Via’s (1991) classic work on GxE and specialization, which we now cite as well. We clarify, too, that these ideas are introduced as hypotheses for future test.

4) Lines 372-373: Again, I appreciate the authors addressing a comment of mine regarding the potential influence of a seed bank, however, presented at this point in the paragraph/discussion it comes across as a throw away sentence, out of place and not contributing much to the paper. I would either further develop this argument a bit, or get rid of it (or move it).

AMS: I can see how this appears as a “throw-away”, but the original point about seed dormancy raised here by the reviewer is a valid one, so I have revised to integrate the point into the general context of how life-history traits may interact in complex ways, thus making inference about underlying causal relationships with fitness difficult.

Reviewer: 2

Comments to the Author(s)

Thank you for the work done in clarifying this manuscript. Unfortunately, I still do not think that the conclusions the authors draw match with their results. I do not think that the GxE shown in this study is sufficient to support temporal GxE as a mechanism for maintaining variation in this system. There is some GxE across these manipulated environments, but this in itself not particularly surprising, and does little to support the main hypothesis of the paper.

Studying these genotypes in more clearly stressed conditions in growth chambers - such as drought or cold stress would provide more interesting results across genotypes and better address the hypothesis than the treatments used in this study.

AMS: First, I agree in principle that we cannot infer directly that fluctuating selection is responsible for the high level of genetic variance observed. However, I wish to point out that the manuscript explicitly makes a claim that is not as strong as the claim being criticized, and is one that is supported by the results. Namely, we claim that a finding of changes in rank-order fitness across environments is consistent with the hypothesis of fluctuating selection as a genetic variance maintaining mechanism. This is by no means a weak claim, but we do not make the stronger claim that fluctuating selection is the only, or even the primary mechanism maintaining genetic variation. I have therefore made an earnest effort to more fully acknowledge and explain limits on inference in several ways. First, I have clarified the benefits and costs of using an observational approach in the Introduction (l.120-121: "...whereas findings consistent with the hypothesis yield a plausible mechanism operating in nature, underlying cause cannot be deduced."), and have fine-tuned the statement of hypotheses with this criticism in mind (l. 134-136), and also in the Introduction have added a full sentence (l.137-139), "Further, we acknowledge that the magnitude and nature of the expression of GxE is necessarily a product of the environments studied, and that the effects might differ under a different regime of environmental variation." Finally, I have also modified the final paragraph to acknowledge the limitation of inference as raised by Reviewer 2: "...because it is the particular qualities and frequencies of sequential environments that will determine the maintenance of genetic variation in nature, quantitative inferences about the importance of temporal fluctuations in maintaining genetic variation cannot be drawn from the present study."

Although I agree in principle with the main criticism of Reviewer #2, I disagree with the more minor point that the approach of using more stressful conditions would have been preferable. Because plasticity expressed across more extreme environments is expected to be greater, such a choice would have made the paper subject to the additional criticism that interpretation of the evidence is limited to the maintenance of genetic variation by GxE only under extreme environmental variation. I have added a short passage on the logic behind the choice of environments upfront in the Introduction (l. 149-151).

Reviewer: 3

Comments to the Author(s)

I think the authors addressed my major comments well in the manuscript. I understand the limitation of sample size for the proposed analyses for this dataset. Please see my specific comments below:

Abstract:

24: meaning of "levels" is not very clear. Do you mean variation is continuous, which could be

as a result of several mechanisms such as f-dependent selection, heterozygote advantage and/or variable selection?

AMS: Yes, “levels” was ambiguous, and I have edited to clarify. This comment alerted me to the same problem two lines later (“levels of genetic variation in *L. inflata*), which I have also edited for clarity.

28: Before *L. inflata* you need a connection word, such as “Thus”.

AMS: Revised to “*L. inflata* thus offers an exceptional...”

39: Before “Although”, the abstract can use a strong statement about the finding of the study.

AMS: True. A slight alteration to has fixed the problem.

Intro:

I like the new Intro a lot. I think the authors addressed my suggestions, thanks for that!

128: I do not think you need the sentences after “Each ...”. Instead give a brief explanation of the results to engage the readers. I think it is ok to discuss the limitations in Discussion only and not mention them in the Intro.

AMS: I appreciate this comment, and agree. The limitations of the study have been retained and revised to satisfy comments of Reviewer #2.

Mat&Met:

150: Fruits or fruit?

AMS: Either seems to be correct. Here, fruit stands for “the fruit from” rather than “all of the fruits of” so we have left this intact.

218: Specifically misspelled.

AMS: corrected.

Results and Discussion:

I think the 2nd and 3rd paragraphs in Discussion read more like Results, and would be better suited for that section. I am sorry this is a vague comment but I think the Discussion needs more work with references to earlier studies. Using the framework constructed in the Intro can guide to talk about why this study agrees/conflicts with some other studies or species would add to the Discussion.

AMS: It is true that there are particular passages that reiterate results; especially the first line of the paragraph. I have thus revised to ensure that the subject is the implications rather than results themselves. The bulk of material in these paragraphs was meant to explore the trends found in the main results, so I have made slight revisions in the spirit of this comment (l. 343-351) to clarify that we are drawing inferences from the results. I agree with the final comment of the need to better place the present results in context of findings of previous studies, and have integrated this context, reflecting the flow of the Introduction, into the first paragraph of the Discussion.

Appendix B

Response to referees

Thank you for your recent decision to accept our manuscript, “Genotype-environment interaction and the maintenance of genetic variation: an empirical study of *Lobelia inflata* (Campanulaceae)” for publication. We especially appreciate your acknowledgment that the comments of the 4th reviewer were obtained because the original critical reviewer was unavailable. The comments are mostly on statistical methods that might have been performed differently. A couple of the main comments (e.g. on population-level analyses; on separating components of GxE) are based either on a misreading of our methods or lack of clarity on our part: our analyses do exactly what the reviewer suggests. However, I have taken all comments seriously and have responded as practical at this stage, as requested.

Reviewer 1: We agree that a sentence stating the main findings is needed near the beginning of the Discussion. I have inserted (l. 305) “Here, we find significant differences in the rank order of genotypic fitness expressed across experimental environments, suggesting the plausibility of GxE as a genetic variance maintaining mechanism.”

Reviewer 4:

“I think that there are lack of results...” Because there are a variable number of genotypes in common among each analysis, we maintain that the best way to illustrate change in rank order is through Figure 1, which contains every genotype used in all analyses. The genotype names are simply labels, and are unimportant. We do present results that include “region” (population) of origin, but the reviewer may have missed this (l. 349) because it is in the Discussion section (along with justification for not including in main results). On the topic of adjusting p-values using FDR, we do this consistently, and have added a phrase to make this clear (l. 242). It is a good point that this procedure increases the probability of a type II error, and we note this alongside relevant results, saying “Although the main effect of genotype is marginally significant according to one of the four analyses—performed on the 16 genotypes present across the two growth chamber environments—this effect is nonsignificant ($P=0.042$) after correcting for multiple comparisons.” However, we disagree that this becomes more important, because failure to detect genotypic effects is relatively unimportant here, and does not alter interpretation of the relative importance of GxE.

“The authors said that they used mixed-effects models...” The reviewer suggests that REML methods might have been more efficient. We opted for separate analyses here—corrected for multiple comparisons—because in our dataset, we not only have imbalance (unequal sample size across analyses), but also variable numbers of subjects (here, genotypes) in common among environments. This is now clarified (l. 225). Indeed Genotype and GxE are random effects, and we are grateful to the reviewer for pointing out what appears to be an error in the F-test; in fact, there is a typo in Table 1 for the MS (MS GxE = 0.0989) rather than an error in the F-test. This has been corrected.

“When analyzing GxE it is possible to separate the two components...” We agree with this comment but are puzzled by it, because we have directly addressed this issue in the manuscript. Indeed, we are specifically interested in the component of GxE due to crossing norms of reaction, rather than to differences in variance among environments. The separation of these components is clearly outlined in our statistical methods (l. 212-219).

We appreciate Reviewer 4’s suggestions for alternatives to Fig. 1. We have tried many of these, including separate graphs for each analysis, and believe that the revised Fig. 1 is the most efficient way to illustrate the GxE effects.